# Effect of Thermal Aging on Viscoelastic Behavior of Thermosetting Polymers under Mechanical and Cyclic Temperature Impact

**DOI:** 10.3390/polym16030391

**Published:** 2024-01-31

**Authors:** Maxim Mishnev, Alexander Korolev, Alexander Zadorin

**Affiliations:** Department of Building Construction and Structures, South Ural State University, Chelyabinsk 454080, Russia; mishnevmv@susu.ru (M.M.); zadorinaa@susu.ru (A.Z.)

**Keywords:** thermal aging, polymer composites, glass fiber-reinforced plastics, thermo-relaxation, representative volume element, viscoelasticity, creep, elasticity, thermal load

## Abstract

Development of load-bearing fiber reinforced plastic (hereinafter referred to as FRP) composite structures in civil engineering, exploited under high temperatures, such as industrial chimneys and gas ducts, requires the knowledge of their long-term behavior under constant and cyclic mechanical and temperature loads. Such conditions mean that the viscoelasticity of FRP should be considered along with the thermal aging effect. This research is devoted to the effects of thermal aging on the viscoelastic behavior of polymers. Two sets of experiments were conducted: creep tensile tests and cyclic heating in a constrained state. The Kelvin–Voigt viscoelasticity model was used to determine the rheological parameters of binder from experimental creep curves. Cyclic heating was used to compare the behavior of normal and thermally aged binders and to evaluate the possibility of temperature stress accumulation. Fourier-transform infrared spectroscopy was used for polymer’s structural changes investigation. Both tests showed that non-aged glassed polymer (hereinafter referred to as GP) was prone to viscoelastic behavior, while the thermally aged GP lost viscosity and worked almost perfectly elastic. It was assumed that long heat treatment had caused changes in the inner structure of the GP, reducing the number of weak bonds and increasing the number of elastic ones. Therefore, the results show that the designing of FRP structures, exploited under thermomechanical load, requires using the elastic model while taking into account the properties of FRP after long-term heat treatment.

## 1. Introduction

Polymer composites are prospective modern materials for civil engineering due to their low weight, high corrosion resistance and high strength [1,2]. They are effective for constructions that are exposed to aggressive environments, e.g., industrial chimneys and gas ducts. Such constructions operate in specific conditions: they are under constant mechanical load for a long period of time (up to decades) and experience high temperatures as well. Thus, the long-term behavior of the material should be determined and considered. In addition, there can be several cycles of heating and cooling, which could lead to the accumulation of temperature stresses. If such stresses do occur and turn out to be significant, they can affect the stress-strain state of the structure [3]. Such stresses, not considered in the design, can complement the stresses from the load and, for example, in total exceed the critical stresses of loss of stability. Therefore, temperature cycles and time must be considered while designing load-bearing polymer constructions.

These days, FRP gas exhaust systems are usually not designed as bearing constructions, and the loads are transferred to supporting constructions (e.g., steel frameworks) [2,3,4]. Using the full material’s capacity could make polymer composite gas exhaust systems cheaper and more reliable. First, if the composite’s bearing capacity was considered, supporting constructions could be made weaker or not used at all. Moreover, the supporting constructions tend to corrode quickly and need repairs, which cost a lot and are sometimes impossible (because it is very expensive and problematic to pause the process of an industrial facility). These days, it is especially relevant as industrial facilities are moving from coal fuel to gas, which results in lower exhaust temperatures and more active condensation and corrosion.

It is known that polymers have a viscous behavior [5,6,7,8] and change their properties under heating [9,10,11]. However, the long-term behavior of polymer composites is not fully researched, and there are no standardized solutions that are needed for civil engineering. Although viscous behavior has been studied in several papers [12,13], the influence of temperature on viscous behavior has yet to be determined. In some papers, the influence of working temperature on stiffness is studied [9,10], but this is for immediate response, and does not consider the viscosity of the material. Although there are papers about the creep of polymers [12,14,15], they use different models and materials, and it is difficult to turn the results into practical solutions. Thermal behavior of polymers and FRPs has also been studied before [11,16] and is researched sufficiently. The research gap lies in how the material works when all these factors are combined: when the construction operates for a long period of time under constant load and changing temperature conditions. To develop the use of load-bearing FRP constructions in civil engineering, we need to study the long-term behavior of polymers, considering rheological behavior and temperature dependencies at the same time.

Previous authors’ research on thermal aging effects showed that the polymer structure right after curing includes several weak bonds with different bond strengths [17,18]. These bonds are responsible for the viscous behavior of the material. As it was stated, thermal aging destroys these weak bonds, leading to an increase in crosslinking and elasticity. Therefore, in this paper, we propose the hypothesis that thermal aging shifts the material behavior from viscoelastic to elastic to some extent. If this is true, thermal aging will become one more factor to be considered when designing polymer constructions as it will directly influence the stress-strain state.

In [19], the authors studied the effect of thermal aging of polylactic acid (hereinafter referred to as PLA) resin on its viscoelastic behavior and performed Fourier-transform infrared spectroscopy (hereinafter referred to as FTIR) analysis. The authors concluded that heat treatment led to structural changes. Although PLA is a thermoplastic polymer, a similar effect is possible for thermoset polymers and this requires investigation. Other scientists studied the thermal aging of thermoset polyurethane [20] and observed the same phenomenon. There was an increase in crosslinking density during the first weeks of heat treatment, followed by degradation of material and chain scission later. Therefore, changes in the structure of polymers can be caused by thermal aging. As the internal structure is responsible for the viscoelastic behavior of material, the effect of thermal aging on viscoelasticity should be investigated.

Viscoelastic behavior can be assessed through such effects as, for example, creep and relaxation. To describe the long-term behavior of polymers and obtain their rheological parameters, mathematical modeling can be used. There are several such models with the same idea of modeling the inner structure of polymers as a combination of primitive elastic and viscous elements, connected in series or parallel [21]. The creep (or relaxation) curves are based on the exponential law. One of the promising models is the Kelvin–Voigt model, which is well-known [21,22,23] and applicable to polymers as well. In this paper, we are going to use the Kelvin–Voigt model (which consists of one elastic and one viscous element connected in parallel) connected to one more elastic element in series. A similar three-element model was described before in classical creep theory [21]. The rheological parameters are acquired through the numerical approximation of theoretical and experimental creep curves.

Thus, the aim of this research is to compare the viscoelastic behavior of material in both normal and thermally aged states and experimentally evaluate the difference between their work in a cyclic heating environment. It includes:–Testing both materials for creep and acquiring creep curves.–Applying the mathematical model of viscoelasticity to obtain rheological parameters and comparing them.–Testing the specimens for cycling heating in constrained conditions and comparing the accumulation of stresses.–Assessing the influence of thermal aging on behavior of material under such conditions.–Performing Fourier-transform infrared spectroscopy to evaluate structural changes caused by thermal aging.

## 2. Materials and Methods

### 2.1. Materials

In this research, we studied the epoxy binder. The same materials as in our previous research were used [11,17,18,24] (presented in Table 1).

The specimens were cured for 2 h at 80 °C and then kept for 24 h at 150 °C. This post-curing was conducted to change the specimen from an extremely plastic under-cured state closer to the real construction material.

### 2.2. Methods

#### 2.2.1. Long Heat Treatment (Thermal Aging)

The same aging conditions were used as in previous studies [18]. The long heat treatment was performed in the oven according to the following program: 168 h (one week) at 160 °C, 168 h at 190 °C, 168 h at 220 °C.

#### 2.2.2. Tensile Testing Chamber

The tests were carried out in a specially manufactured chamber that allowed the ends of the sample to be pinched from longitudinal displacements and to test it for tension by moving one of the chamber parts. Heating elements and a fan were installed in the chamber to create hot air movement and uniformity of heating inside the chamber.

The chamber was installed on a Tinius Olsen h100ku machine (Tinius Olsen Ltd., Surrey, England) (characteristics were presented in previous studies [11,18,24]). The upper part of the camera (see Figure 1) was clamped in the grip of the machine and is able to slip freely relative to the lower part. Clamps had flexibility for turns, so bending moments from eccentricities did not occur in the sample, and it worked only for longitudinal force.

The strain of the sample was controlled by clock-type sensors to eliminate the effect of slippage in the grips of the machine and the test chamber. Four sensors with an accuracy of 0.01 mm were used, mounted in pairs crosswise (to account for possible slopes relative to both axes of symmetry of the sample section).

To increase the accuracy of determining the calculated length, the contact of the corner shelf with the sample along its length was minimized by inserting the spacer and making sure that the contact between the corner and the specimen was concentrated in one spot. The test scheme and the photo of the real installation are presented in Figure 2.

#### 2.2.3. Testing for Creep

The Tinius testing machine allowed us to make a program for constant loading. The machine automatically increases the strain if the tensile force falls. Consequently, it was used for creep test as it represents the conditions for creep: constant stress and unconstrained displacement.

The sensor readings were recorded at zero force immediately after applying load and then with the steps of 1, 5, 10, 15, 30, and 60 min. They were transformed into span change using Formulas (1)–(7) (see Figure 3). Then, the strain ε was calculated by dividing the displacement by the initial span. As a result, creep curves ε(t) were obtained (t—time).
(1)∆L=∆1−∆2 
(2)$$d1^{\prime }=d1-∆1+D$$
(3)$$∆1=d1+D-d1^{\prime }$$
(4)$$d2^{\prime }=d2-∆2+D$$
(5)$$∆2=d2+D-d2^{\prime }$$
(6)$$∆L=d1+D-d1^{\prime }-d2-D+d2^{\prime }$$
(7)$$∆L=d1-d1^{\prime }+d2^{\prime }-d2$$
where readings d1 and d2 are averaged for two crosswise sensors for near and far sections, respectively.

#### 2.2.4. Acquiring Rheological Parameters

In this research, we used the three-element model, which is based on the Kelvin–Voigt model. This model is described in classical creep theory [21]. The appearance of the model is shown in Figure 4.

$E_{1}$ and $E_{2}$ are Young’s modulus for elastic elements. K is a parameter of viscosity of viscous element and $\sigma_{1},\;\sigma_{2},\;\sigma_{K},\;\sigma$ are normal stresses in elastic elements, viscous element and the material overall, respectively. These parameters are connected by the following relations [21] (Formulas (8) and (9)):(8)$$H=\frac{E_{1}\cdot E_{2}}{E_{1}+E_{2}}$$
(9)$$n=\frac{K}{E_{1}+E_{2}}$$

H is a long-term elastic modulus, n is a relaxation time.

The creep law is described by (10):(10)$$\varepsilon \left(t\right)=\frac{\sigma }{H}+\sigma \left(\frac{1}{E}-\frac{1}{H}\right)e^{-H\cdot \frac{t}{En}}$$

Experimental creep curves were imported into the Mathcad program, which created a system of equations ε(t) (see (10)), where strain, time and elastic modulus are known parameters and H, n are unknown. The elastic modulus was obtained in previous research [11,24] and checked again in this one at initial loading. By equalizing the theoretical and experimental strain values at given times, the program numerically approximated them and came up with H, n, which provided convergence of theoretical and experimental curves.

As $E_{1}=E$ [21], there are two unknown parameters remaining: $E_{2}$ and K. By solving the system of Equations (8) and (9), they were defined.

As a result, all parameters of the model were determined.

#### 2.2.5. Cyclic Heating Test

The specimen was clamped on both sides and pre-stressed for a load of 1000 N. This was conducted to get rid of possible shifts in installation. After the initial load, the specimen was left to rest for 5 min to let the initial fast relaxation slow down and not disturb the results.

The temperature was automatically controlled by a thermostat and checked according to a reference sample of the same thickness as the test sample with a thermocouple glued inside.

The testing process consisted of turning on the heating until the desired temperature was reached, turning the heating off until the specimen cooled down to room temperature and then repeating the cycle several times.

The measurements were recorded every 5 min during the whole testing period. The temperature was recorded from the reference sample, and the variation from the thermostat was no more than 2 °C.

From previous research [11], the dependencies of the modulus of elasticity (E) and coefficient of thermal expansion (CTE) from temperature are known. Previously, the relationship between E and CTE was established, and it was found that, up to a temperature of about 80 °C, they change weakly and almost linearly. After 80 °C, there is a sharp change in the behavior of the material and a non-linear growth of E and CTE. At this stage of research, it was assumed that the viscoelastic behavior of the material would change similarly, since all these processes are related to the internal structure of the material. Since the determination of rheological parameters was carried out at normal temperature (see further), heating tests were carried out at 70 °C in order to exclude the possible influence of non-linearities that have not yet been studied.

#### 2.2.6. Fourier-Transform Infrared Spectroscopy

In order to verify the changes in the chemical structure of EP-TR during the long heat treatment process, FTIR was conducted. FTIR [25] is a technique used to obtain an infrared spectrum of absorption or emission of a solid, liquid or gas. The goal of absorption spectroscopy techniques (FTIR, ultraviolet-visible (“UV-vis”) spectroscopy, etc.) is to measure how much infrared (hereinafter referred to as IR) radiation a sample absorbs at each wavelength.

In this research, FTIR spectrophotometer IRAffinity-1S was used (Figure 5). The manufacturer is Shimadzu, Kyoto, Japan. IRAffinity-1S is a spectrometer with wavenumber range from 7800 to 350 cm^−1^ and a spectral resolution from 0.5 to 16 cm^−1^.

## 3. Results

### 3.1. Creep Tests

To assess the correctness of measured strains, the Young’s modulus was calculated using Hooke’s law. The modulus was 3000 MPa±10%, which corresponds to our previous research [11,24] and other scientists’ data [9]. Thus, the experimental data was considered to be correct.

The testing equipment did not allow us to test both EP and EP-TR samples under the same stress. However, the rheological parameters should not depend on stress, which was confirmed in this research (see Section 3.2). The shape of the curves for both stress (σ) levels (see Figure 6) was the same. Strain (ε), as expected, was two times higher for the doubled stress.

As we can see in Figure 6 and Figure 7, the behavior of normal and thermally relaxed binder under constant load is different. EP samples gave us classical creep curves, with strain increasing over time and gradually slowing down. On the other hand, EP-TR showed almost no viscous behavior. Strain-time dependency is completely different when compared to EP. There is a rapid increase in strain in the beginning (which was followed by audible crackling) and then the strain change is negligible. During the first hour, the strain change of EP was 3%, while for EP-TR the change was 0.2% (excluding the initial shift and considering only the region that corresponds to creep). In summary, EP-TR worked like a brittle material, prone to cracking and not having viscous behavior.

### 3.2. Obtaining Rheological Parameters

First, we used the model to calculate the parameters H, n of EP for both stress levels. The average results are presented in Table 2.

The results for both stress levels were quite close. This was expected, as the parameters in such models as the one we used should not depend on stress level. These parameters represent the inner structure of the material as well as the Young’s modulus. The results seem to be adequate.

The rheological parameters of model’s elements were calculated for average values from Table 2 and are shown in Table 3.

Using the parameters, theoretical creep curves were created and compared to experimental ones. The result is shown in Figure 8.

Indeed, the calculated rheological parameters represent the real material behavior.

The biggest error is at the end of the curve for 22.64 MPa. The experimental strain value here is 0.00859, and the theoretical one is 0.00867. Thus, the error is 0.93%, which is satisfying.

The situation is different for EP-TR. As was mentioned before, the aged binder became non-viscous and brittle. Its strain-time curve (Figure 7) does not correspond to the law (Formula (10)) and calculating rheological properties was unreasonable.

### 3.3. Cyclic Heating

The time-temperature-force dependencies are presented in Figure 9 and Figure 10 for EP and EP-TR, respectively.

As the figures show, the temperature stress accumulation was much lower in EP-TR. By the end of third cycle, EP had lost 11.6% of tensile force, while EP-TR had lost 6.5%.

By the end of the shown cycles (five for EP and three for TR), there was almost no further difference.

The minimum force for EP was 25.5% lower at the end, and 9.7% lower for EP-TR.

In both cases, slower or faster, we see that over some cycles the changes become negligible. The viscoelastic model we used earlier has a special feature: as can be seen from the Formula (10), the creep slows down over time, stopping at some point:(11)$$\varepsilon \left(t\to \infty \right)\to \frac{\sigma }{H}$$

It also proves the applicability of the selected model for the studied material. This feature can be used in future to make construction calculations easier, as the maximum creep strain could be defined from (11).

### 3.4. Fourier-Transform Infrared Spectroscopy

According to the results of FTIR, it was determined that heat treatment led to a 1.5–2-fold increase in the absorption of IR waves in all spectra. The results are shown in Figure 11.

## 4. Discussion

Summarizing the results, three separate experiments have indicated that thermal aging had caused some changes in GP.

According to results, a non-aged polymer (EP) has:(1)Viscoelastic behavior under mechanical load that corresponds to the Kelvin–Voigt model;(2)Significant stress accumulation during cyclic heating impact. This indicates viscoelastic behavior;(3)Low IR-radiation absorption in the main spectra, as shown by the FTIR analysis.

The aged polymer (EP-TR) has:(1)Absence of viscosity in its behavior. During creep tests, the strain change over time was negligible (0.2%). The initial strain change was, presumably, caused by the appearance of inner cracks. Therefore, the EP-TR’s stress-strain state cannot be described using creep theory, as such material does not show viscous behavior. It has both advantages and disadvantages. On the one hand, elastic behavior is good for calculations as it can be easily determined and does not have such effects as stress accumulation. On the other hand, brittleness is a negative factor for constructions. Cracks have to be considered in calculating stiffness (in analogy with concrete) and the destruction is immediate and dangerous;(2)Increased elastic behavior during cyclic heating tests. The change in longitudinal force during cycles is the sum of accumulated temperature compression stress and relaxation, accelerated by high temperature. Both are connected to viscosity. Therefore, a smaller change means less viscosity. As we can see (Figure 10), EP-TR showed significantly smaller changes in longitudinal force during cyclic heating tests. Hence, it behaves more elastically than EP. As a result, the data from creep tests, which showed the decrease in viscosity, correlates with cyclic heating results;(3)An increase in IR-radiation absorption during FTIR analysis up to two times.

It is known that viscoelastic behavior of polymers is connected to their inner structure [8,17]. The difference in behavior of thermally aged and non-aged GP under constant mechanical and cyclic thermal loads might be explained by the changes in the structure of the material, which were caused by long-term heat treatment. As weak volatile bonds are destroyed during thermal aging, the material loses its ability for plastic deformation, becoming elastic and brittle [17].

The verification of structural changes was performed using FTIR analysis. As FTIR analysis showed, long heat treatment has resulted in a decrease in IR waves transmission. It can be explained by less free space in the inner structure, which was caused by reduction of free polymer groups and more crosslinking.

Other researchers have got similar results. In [19] the authors studied the effect of thermal aging and performed FTIR analysis. FTIR tests have shown an increase of absorbance after heat treatment. The authors observed and explained the changes in chemical structure of the material, concluding that there was an increment in crosslinking density during physical rearrangement. They concluded that heat treatment leads to more stable structures through structural changes. In another research, thermal aging of thermoset polyurethane was studied [20]. Based on the results, the authors concluded that during the first weeks of aging, there was an increase in crosslinking density.

Thus, as was proven by current results and other research, long-term heat treatment results in structural changes of material. These changes are supposed to shift the behavior of material from viscoelastic to elastic and two separate experiments (creep tests and cyclic heating tests) indicated this phenomenon to exist.

Therefore, the hypothesis, which was mentioned in the Introduction (that thermal aging decreases viscosity), is supported by the results of three independent experiments and correlates with other research [19,20].

## 5. Conclusions

In this study, the following goals were achieved:

-EP and EP-TR binders were tested for creep, and $\varepsilon \left(t\right)$ curves were obtained.-The mathematical Kelvin–Voigt model of viscoelasticity was successfully used to get the rheological parameters of EP. The EP-TR turned out to lack viscous behavior and was brittle. The coefficient of this EP model is determined.
$$K=\;1.53\;\times \;10^{8}\;\mathrm{MPa}\cdot s.$$-The long-term behavior of EP and EP-TR was compared. Both binder variants were tested for cyclic heating in restrained conditions and their behavior was also compared.-The influence of thermal aging on viscoelastic behavior was assessed. The result of thermal ageing is the loss of creeping under the load and decreasing of stress accumulation under cycling loading.-The observed shift in material behavior was explained by structural changes, and these changes were verified by FTIR analysis.

The non-aged epoxy binder, as expected, shows viscoelastic behavior, which can be described using the three-element Kelvin–Voigt model. It showed similar material constants for two different stress levels; the behavior of theoretical strain-time creep curves corresponded to the experimental curve. Future research will be based on this model.

Thermal aging, as was previously stated in our previous research [17,18] and by other authors [16,19,20,26,27], influences the properties of material. Based on this knowledge, a new hypothesis was proposed: aging also influences viscoelasticity. This hypothesis was proved by two independent experiments: creep testing and cyclic heating. Indeed, EP-TR appeared to behave non-viscously in creep test and almost elastically in cyclic heating. This change can be explained by changes in the inner structure of material: long heat treatment results in increased crosslinking and reduced number of weak bonds. The proposed explanation was verified by FTIR analysis: an aged specimen transmits less IR-radiation, which means that its structure became denser.

Hence, this research shows that thermal aging resulted in changes in the inner structure of thermoset epoxy polymer binder. Consequently, the material behavior had also changed: it became less viscous and worked almost perfectly elastically during experiments.

Thermal aging should be investigated more, as it not only has positive effects, but also increases brittleness and crack formation.

Stress accumulation from cyclic heating and cooling directly influences the stress-strain state of construction and should be considered in the future when designing such constructions. Depending on the temperature regime, the relaxation or accumulation can be up to 25% of stress (based on current research data). It cannot be ignored while designing load-bearing polymer composite constructions and requires more research to get safety coefficients to account for these phenomena. Also, it is necessary to consider the transition from viscoelastic to elastic and fragile structure of FRP during long term heating exploitation while designing such constructions. Therefore, further research on viscoelasticity and thermal behavior is needed.

## Figures and Tables

**Figure 1 polymers-16-00391-f001:**
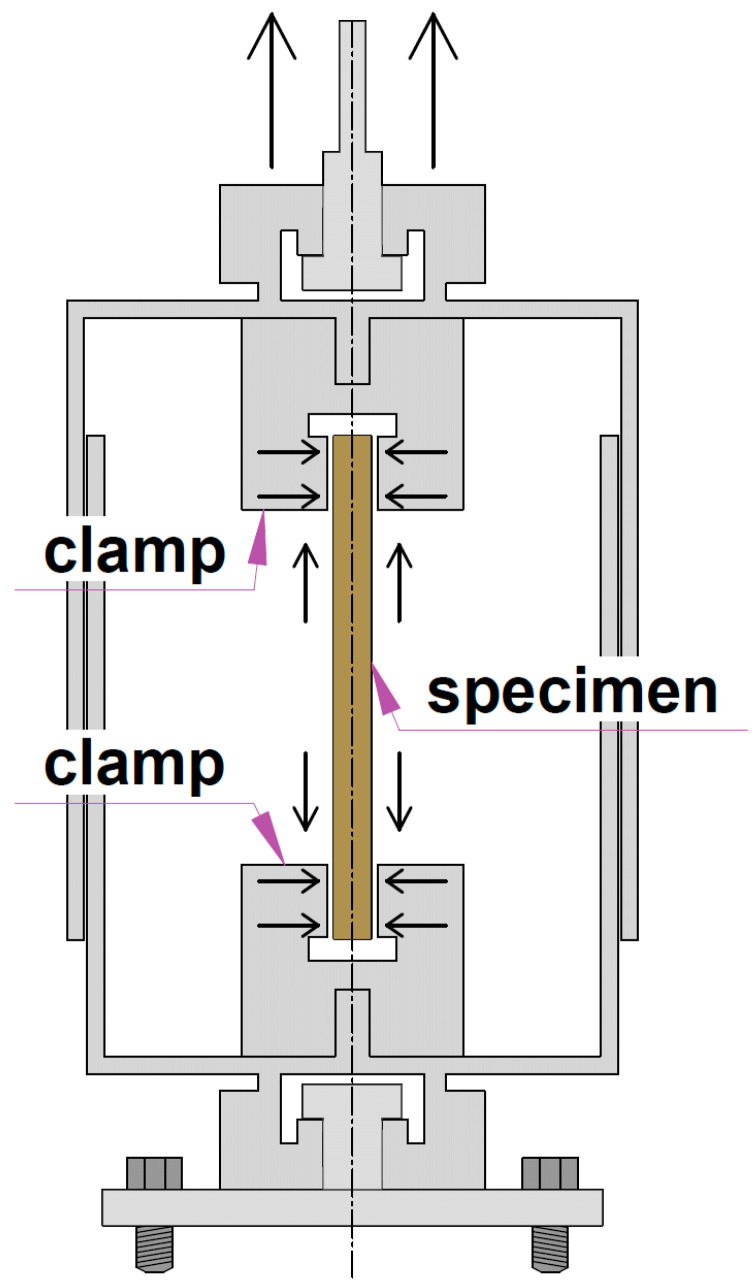
Testing chamber.

**Figure 2 polymers-16-00391-f002:**
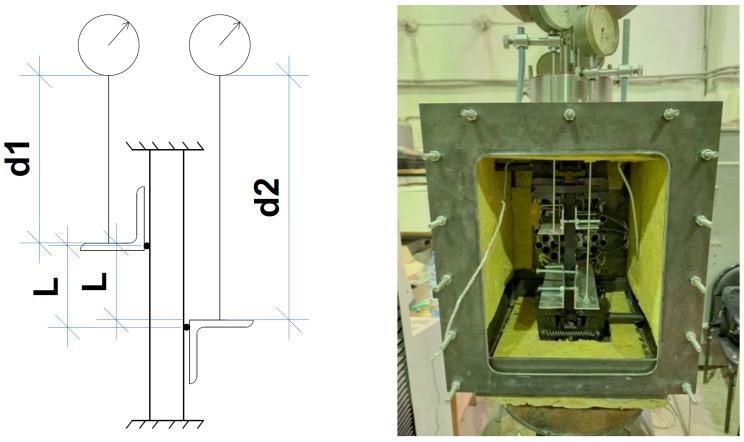
The test scheme (**left**) and the real photo (**right**).

**Figure 3 polymers-16-00391-f003:**
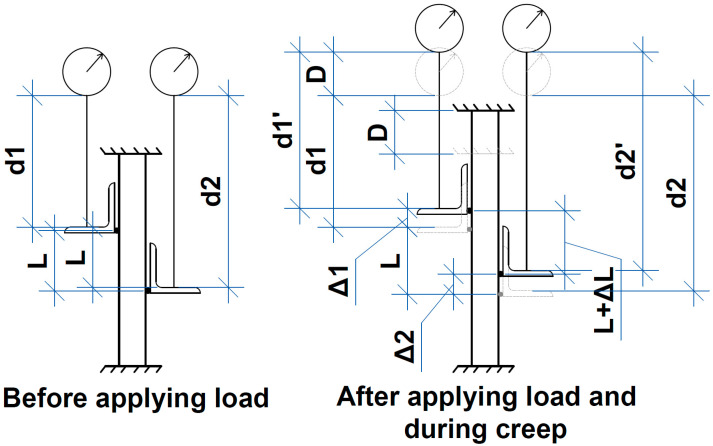
The scheme for Formulas (1)–(7).

**Figure 4 polymers-16-00391-f004:**
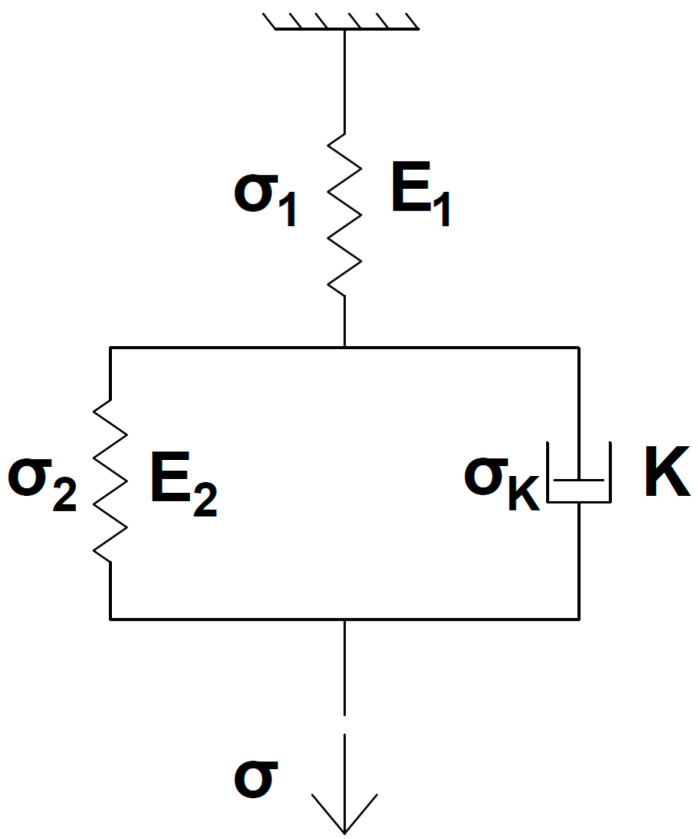
Three-element viscoelastic model.

**Figure 5 polymers-16-00391-f005:**
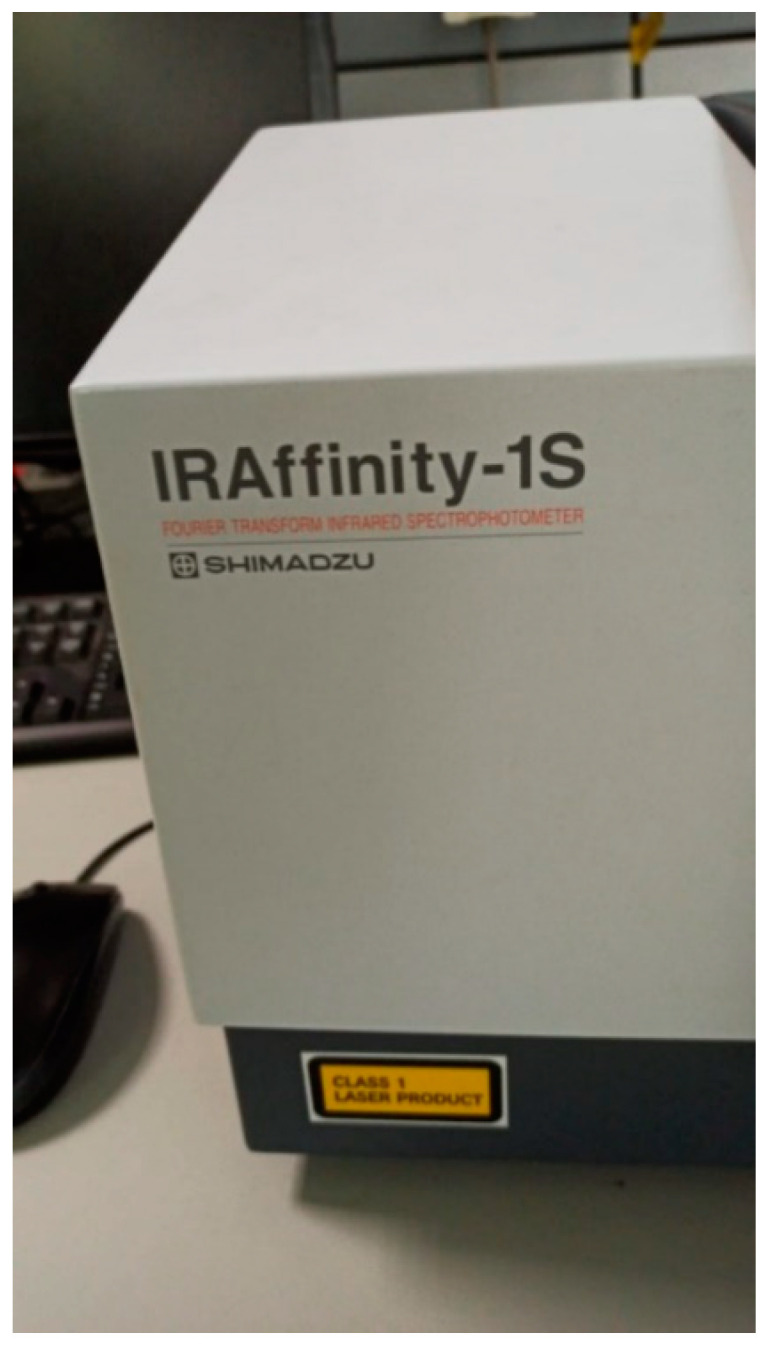
FTIR spectrophotometer IRAffinity-1S.

**Figure 6 polymers-16-00391-f006:**
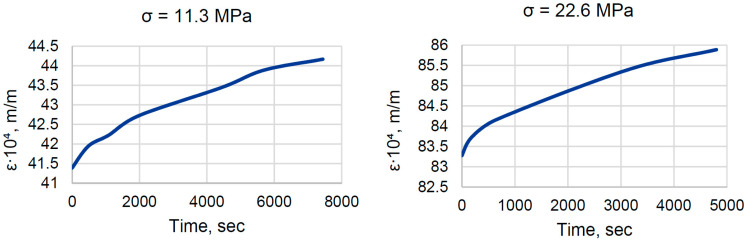
Creep curves for EP.

**Figure 7 polymers-16-00391-f007:**
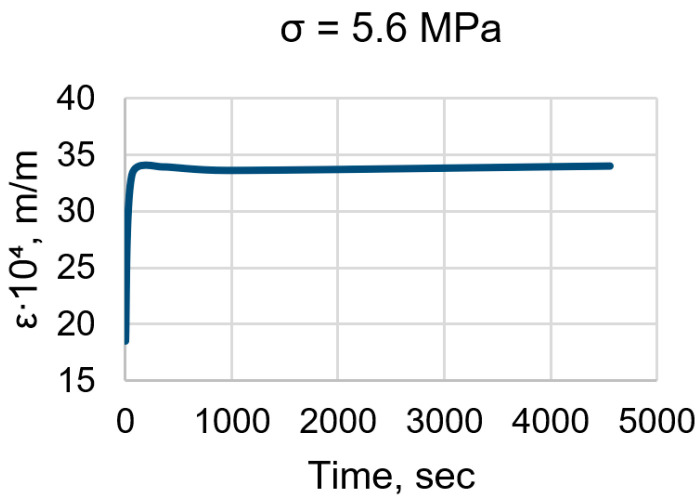
Strain-time curve for EP-TR.

**Figure 8 polymers-16-00391-f008:**
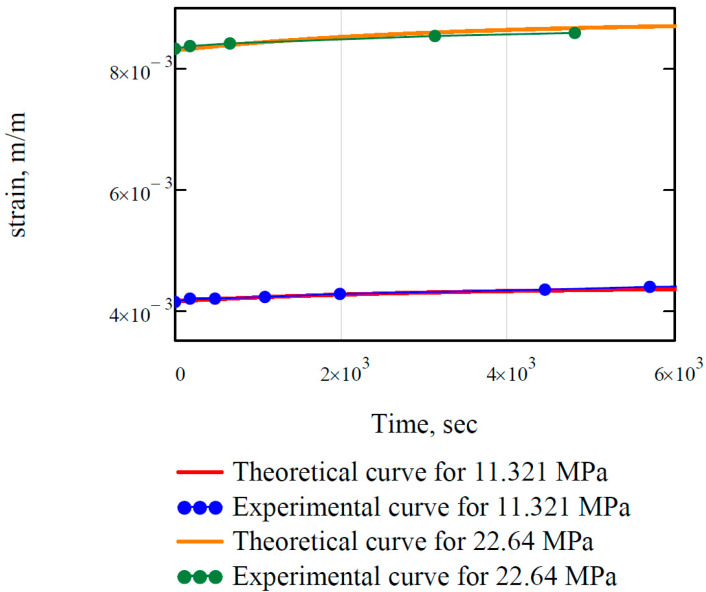
Comparing theoretical and experimental creep curves for EP.

**Figure 9 polymers-16-00391-f009:**
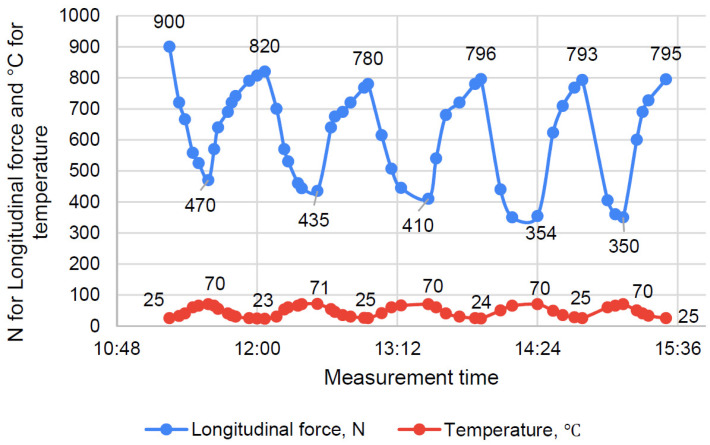
Cyclic heating of EP.

**Figure 10 polymers-16-00391-f010:**
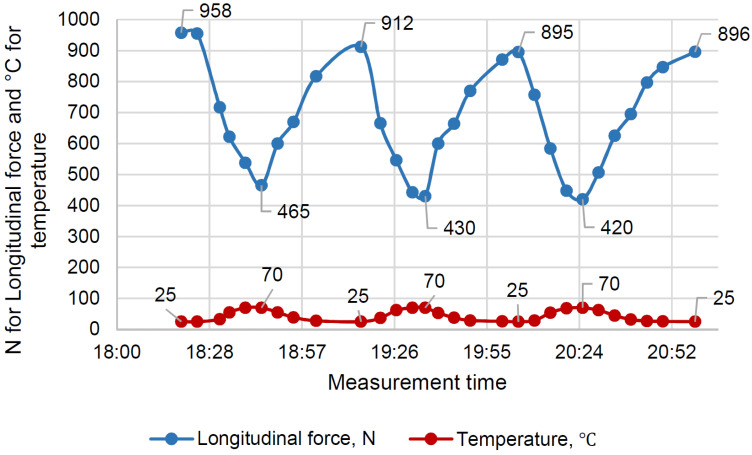
Cyclic heating of EP-TR.

**Figure 11 polymers-16-00391-f011:**
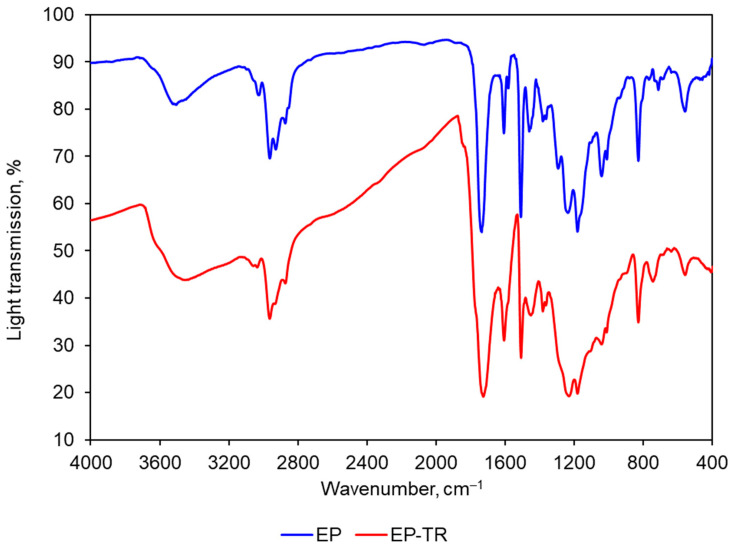
FTIR analysis results.

**Table 1 polymers-16-00391-t001:** Types of binders investigated.

№	Composition	Name
1	Epoxy (Ker 828 52.5% + MTHPA 44.5% + alkofen 3%) (without thermal aging)	EP
2	Epoxy (Ker 828 52.5% + MTHPA 44.5% + alkofen 3%) (thermally aged, see Section 2.2.1)	EP-TR

**Table 2 polymers-16-00391-t002:** Parameters for EP.

Composition	*σ*, MPa	*E*, MPa	*H*, MPa	*n*, min
EP	11.32	2735	2550	51.4
	22.64	2719	2620	48.7

**Table 3 polymers-16-00391-t003:** Element’s parameters for EP.

Composition	*E*_1_, MPa	*E*_2_, MPa	*K*, MPa·s
EP	2727	4.8 × 10^4^	1.53 × 10^8^

## Data Availability

Data are contained within the article.

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
