# Peer review of "Effect of Thermal Aging on Viscoelastic Behavior of Thermosetting Polymers under Mechanical and Cyclic Temperature Impact"

_polymers, 2024, doi:10.3390/polym16030391_

Round 1
Reviewer 1 Report
Comments and Suggestions for Authors
The manuscript entitled "Effect of Thermal Aging on Viscoelastic Behavior of Thermosetting Polymers Under Mechanical and Cyclic Temperature Impact" deals with the characterization of the viscoelastic behavior of an epoxy sample, before and after a thermal aging treatment. The topic object of the work should be of some interest. However, the manuscript needs to be enriched before publication. In particular, some further characterizations and a more deep discussion about the phenomena occurring during the thermal aging must be introduced, as well as a more extensive discussion about the obtained results.
Other remarks:
- Please, explicitate FRP in the introduction.
- The conditions of the thermal aging treatment must be reported in section 2.2.1.
- The quality of the Figures should be enhanced in order to improve their readability.
- Concerning Figures 5-6, explain better the obtained curves and the differences observed as a function of the applied load. In the text, please correct "Figures 4 and 5" and also check the title of Figure 6.
- Concerning the results depicted in Figure 6, the Authors attributed the different behavior of the aged sample to the cleavage of weak chemical bonds occurring during the thermal treatment. Further analyses (such as FTIR) are needed to verify the inferred modifications of the chemical structure of the material.
- Please, explain better the meaning of the sentence "There-fore, the is no rheology in EP-TR’s behavior" in section 3.2.
- y axes of Figures 8 and 9 should have the same scale, to allow a more easy comparison between the two set of data.
- The supposed influence of aging on the sample viscoelasticity should be deeper discussed and verified. In which way the thermal aging did affect the viscoelastic behavior of the material? Which are the underlying phenomena causing the modification of the viscoselastic behavior? The Authors should discuss the modification of the material structure causing a variation of the viscoselastic behavior, also referring to some results already reported in the literature.
Comments on the Quality of English LanguagePlease, check the whole manuscript in order to correct the typos.
Author Response
Thank you very much for taking the time to review this manuscript. Please find the detailed responses in the file below and the corresponding corrections highlighted and tracked in the re-submitted file of manuscript.
Please see the attachment.

Reviewer 2 Report
Comments and Suggestions for Authors
This article mainly studies the effect of thermal aging on the viscoelasticity of polymers, which has certain guiding significance for the application of polymers under high temperature and fixed mechanical loads. But there are still some issues:
1.The logic of the abstract is not very clear, and the most important conclusion is not highlighted.
2. The image format of the text is not consistent.
3. The author's explanation of the experimental results is insufficient. A more detailed explanation of the experimental results and why these conclusions were drawn should be provided.
4. The conclusion of the article is not clear. What important results or findings should the author emphasize?
Author Response

(The authors gave the same response as above.)

Round 2
Reviewer 1 Report
Comments and Suggestions for Authors
I recommend the publication of the manuscript as it stands, as the Authors revised it following the Reviewer' suggestions.
Reviewer 2 Report
Comments and Suggestions for Authors
This article mainly studies the effect of thermal aging on the viscoelasticity of polymers, which has certain guiding significance for the application of polymers under high temperature and fixed mechanical loads. But the problem page is very obvious:
1. Abbreviations such as "FRP" that appear for the first time are not labeled with their full names;
2. In the introduction section, some large paragraphs lack references, while others have ten references at the end of a sentence, which is unreasonable;
3. The logical and overall coherence of the abstract is poor;
4. It is recommended to adjust and optimize the image format in the text;
5. The explanation of the experimental results is not sufficient. You should provide a more detailed explanation of your experimental results and why you came to these conclusions;
6.The conclusion of the article is not clear enough, you should emphasize your important findings.
Comments on the Quality of English LanguageThe symbols in the main text are inconsistent with those in the charts, and there are errors in punctuation. Please make careful modifications.
Author Response

(The authors gave the same response as above.)

Round 3
Reviewer 2 Report
Comments and Suggestions for Authors
It is recommended that authors provide more convincing images, data, and other materials in their future work. Although the current work appears to be relatively complete, it is a judgment of trust in the authors.